# MTMT: Tiered Treatment Effect Decomposition for Multi-Task Uplift Modeling

## Abstract

As a key component in boosting online user growth, uplift modeling aims to measure individual user responses (e.g., whether to play the game) to various treatments, such as gaming bonuses, thereby enhancing business outcomes. However, previous research typically considers a single-task, single-treatment setting, where only one treatment exists and the overall treatment effect is measured by a single type of user response. In this paper, we propose a Multi-Treatment Multi-Task (MTMT) uplift network to estimate treatment effects in a multi-task scenario. We identify the multi-treatment problem as a causal inference problem with a tiered response, comprising a base effect (from offering a treatment) and an incremental effect (from offering a specific type of treatment), where the base effect can be numerically much larger than the incremental effect. Specifically, MTMT separately encodes user features and treatments. The user feature encoder uses a multi-gate mixture of experts (MMOE) network to encode relevant user features, explicitly learning inter-task relations. The resultant embeddings are used to measure natural responses per task. Furthermore, we introduce a user-treatment feature interaction module to model correlations between each treatment and user feature. Consequently, we separately measure the base and incremental treatment effect for each task based on the produced treatment-aware representations. Experimental results based on an offline public dataset and an online proprietary dataset demonstrate the effectiveness of MTMT in single/multi-treatment and single/multi-task settings. Additionally, MTMT has been deployed in our gaming platform to improve user experience.

## 1 Introduction

To offer a better personalized experience and increase user engagement, online marketing platforms usually provide incentives such as advertisements Lo (2002), discounts Gubela et al. (2017), and bonuses Ai et al. (2022). Although these incentives are crucial for generating additional revenue and activity, they are often costly, and individual users can have varied responses to different incentives. For example, some users will not play the next game without a bonus, while others will continue to play regardless. Consequently, accurately modeling individual users' responses and identifying the target user groups that are likely to be positively affected by incentives is essential for enhancing marketing benefits Xu et al. (2022).

One of the fundamental challenges to measuring the response is the existence of the counterfactual problem, where an individual is either treated (treatment group) or not treated (control group). Therefore, in the same context, we can not simultaneously observe a user's response to a certain incentive or no incentive. Such a problem can be referred to as causal inference Yao et al. (2021). To resolve this, uplift modeling Gutierrez & Gérardy (2017) has been proposed to estimate the individual treatment effect (ITE) (a.k.a. uplift) that describes how individual user responds to an incentive Zhang et al. (2021).

The current uplift modeling frameworks predominantly concentrate on directly modeling the response functions of both treatment and control groups to infer counterfactual predictions. Among them, meta-learner-based methods leverage existing models to estimate the Individual Treatment Effect (ITE) of personalized treatments. For example, S-learner Künzel et al. (2019) estimates the conditional average outcome of

treatment and control group, then calculates ITE through subtraction. Building on this, other two-step meta-learners were proposed with other additional operations, including X-Learner Künzel et al. (2019), DR-Learner Kennedy (2023), R-Learner Nie & Wager (2021), etc. Nevertheless, these methods are prone to be influenced by the sample imbalance between the treatment and control groups. Another line of work involves tree-based models, which divide the user population into sub-groups according to specific splitting criteria (e.g., sensitivity to the treatment) and predict the uplift on each leaf node. A notable example is the causal forest Athey & Imbens (2016), which integrates multiple trees to estimate heterogeneous treatment effects. With advances in deep learning, numerous neural network-based models have been developed that learn embeddings from related features, thereby predicting the uplift more flexibly. Based on representation learning, such models either predict the treatment effects of treatment and control groups separately Shi et al. (2019); Shalit et al. (2017); Curth & Van der Schaar (2021a;b); Schwab et al. (2020); Zhong et al. (2022), or directly models uplifts from user, context, and treatment features Ke et al. (2021); Liu et al. (2023); Huang et al. (2024). In this work, we focus on neural network-based models and propose to directly model uplifts.

While prevailing uplift models perform adequately on synthetic and product datasets, they exhibit two notable limitations. First, in real-world scenarios, multiple treatments often impact the target response, and multiple responses contribute to overall outcomes (multi-treatment multi-task). However, most models focus on a single treatment and a single target response (single-treatment single-task), overlooking the complex interactions between treatments and responses. This simplification can result in incomplete representations and biased ITE estimates. Several representation-based approaches address uplift modeling in multi-treatment or multi-task settings. Addressing the multi-task problem, Huang et al. (2024) studied two chained tasks (click-through rate and click-conversion rate) by designing a two-branch encoder with shared parameters between branches to encode features, subsequently outputting uplift scores for each task. For the multi-treatment setting, Sun & Chen (2024); Velasco-Regulez & Cerquides (2023) designed multi-head networks that predict the natural response (control group) and responses from multiple treatments, then calculate uplifts through subtraction. Additionally, Liu et al. (2023) considered applied separate encoders to encode treatment and user features, combining them to learn a unified representation, and then calculating the uplifts for each treatment. However, they assume the equivalence of different treatments and learn a shared representation across multiple groups. Nevertheless, in online applications, the response difference among treatments can be much less notable than the response difference between treatment and control groups. We illustrate this point in Fig. 1. Therefore, simply combining the treatments and estimating the uplift can lead to suboptimal estimation.

Secondly, existing models commonly focus on utilizing user and contextual features while neglecting the treatments. However, the correlation between treatments and user profiles is crucial for uplift modeling, especially in a multi-treatment setting. For instance, we observe that low-active users are more likely to play the next game when given a higher-valued bonus, whereas high-active users may continue to play the game regardless of receiving a bonus. While several works have incorporated treatment features as input to enhance the accuracy of estimation, they fall short in modeling the interaction among treatments and user features in a task-oriented way. For example, Liu et al. (2023); Huang et al. (2024); Ke et al. (2021); Xu et al. (2022) separately encoded the treatment features and non-treatment features, then generated a unified embedding either by concatenating or by weighted addition. This operation indirectly combines the treatments with other features but overlooks the implicit relationship between treatments and various tasks.

To address the aforementioned limitations, we propose a Multi-Treatment Multi-Task (MTMT) uplift modeling framework that directly models the user-treatment interactions. We follow a divide-and-conquer method by decomposing multi-treatment into two tiers, where the base treatment defines whether a user receives treatments, and the secondary treatments define the specific types and amounts. For instance, one must first decide whether to give a user a game bonus (which we refer to as *Treatment Decision*) and then decide which bonus to give (which we refer to as *Treatment Selection*). Following this, we separately estimate the base uplift for the treatment group and the incremental uplifts on top of the base uplift for the specific treatments. MTMT encodes the user features through a representation network based on a multi-gate mixture-of-experts. The generated embeddings are projected to compute the natural response for each task. Meanwhile, treatment encoders are employed to independently encode the base treatment and its subsequent

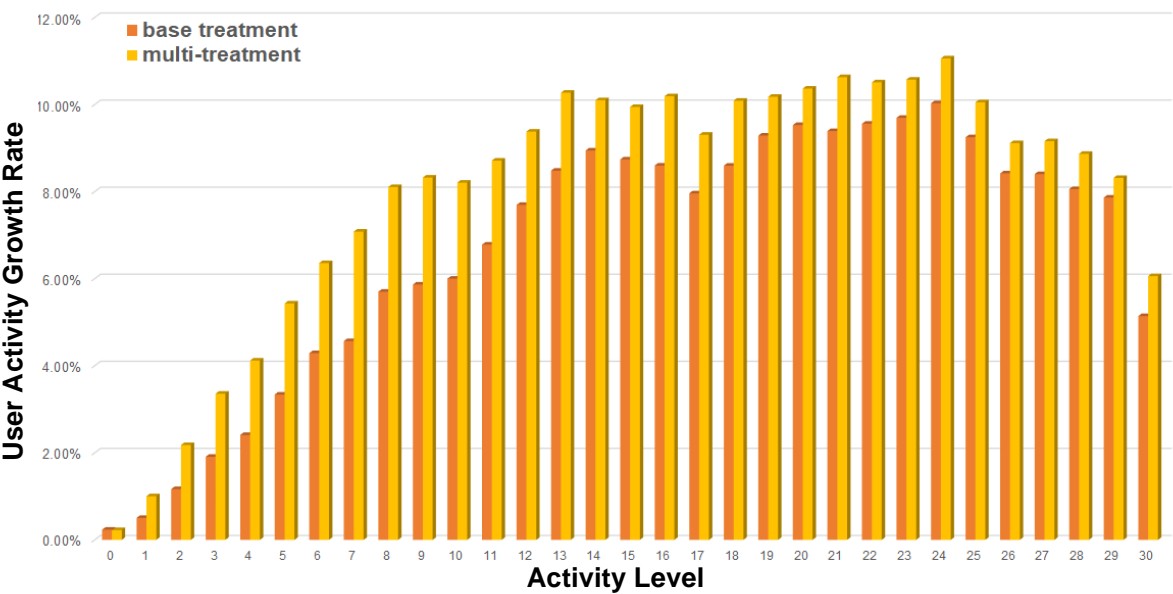

Figure 1: Activity growth rates by base treatment and multi-treatment deployment on our online gaming server. Growth rates are shown separately based on users' historical activity, as past activity significantly impacts future activity.

secondary treatment information. Next, a user-treatment interaction module explicitly models how treatments attend to each user feature. The combined information is further enhanced and projected to predict the response of different treatment groups. This approach ensures a more accurate and nuanced estimation of treatment effects, addressing the complexities of multi-treatment and multi-task scenarios. In summary, our contributions are as follows:

- **Background**: We aim to accurately model the uplifts with multiple potential treatments and multiple target responses. To the best of our knowledge, this is the first effort in multi-treatment multi-task uplift modeling without underlying assumptions about the treatments or tasks.

- **Method**: We introduce a novel uplift model that explicitly captures user-treatment feature interaction through the self-attention mechanism. By separately estimating the base uplifts and the uplift differences between treatments, the model accounts for the minimal uplift variance among treatments, thereby accurately estimating uplifts of different treatments and tasks in an end-to-end manner.

- **Evaluation**: We demonstrate the effectiveness of the proposed model using a public dataset and a large-scale product dataset with multiple tasks and treatments. Our results indicate that MTMT significantly outperforms its competitors. The MTMT model has been deployed on our online gaming platform, serving millions of users.

## 2 Related Works

### 2.1 Uplift Modeling

Uplift modeling aims to establish a difference in the users' behaviors when applying or not applying certain treatments by measuring the corresponding ITE. Existing uplift research mainly focuses on three settings: 1) single-treatment single-task setting. For example, meta-learner-based methods Künzel et al. (2019); Kennedy (2023); Nie & Wager (2021) integrate existing models to predict ITE. Besides this, tree-based methods Wager & Athey (2018); Athey & Imbens (2016) gradually divide subpopulations by different metrics and estimate the ITE at the leaf node. Due to their superior feature extraction abilities, deep neural networks have gained

much popularity in uplift modeling. They either directly model the uplift from learned representations Ke et al. (2021), or separately estimate the natural response and treated response, then calculate the uplift by subtracting Shalit et al. (2017); Shi et al. (2019); Curth & Van der Schaar (2021a); Gutierrez & Gérardy (2017); Schwab et al. (2020); Zhong et al. (2022). 2) single-treatment multi-task setting. Huang et al. Huang et al. (2024) interpret the user behaviors as a sequential chain, and include two chained tasks to estimate their uplifts. They first extract user and treatment features with an encoder, then design a network with two branches that have shared parameters to predict the uplifts of the two tasks. 3) multi-treatment single-tasks setting. Zhao et al. Zhao & Harinen (2019); Zhou et al. (2022) extend several meta-learners to the multi-treatment setting. Other works adapt tree-based methods to estimate multiple treatments' uplifts Zhao et al. (2017b;a). For the neural network-based approach, Sun & Chen (2024); Velasco-Regulez & Cerquides (2023); Mondal et al. (2022) design independent heads to estimate each treatment group's response and the control group's response. Liu et al. Liu et al. (2023) employ the treatments as the input and explicitly extract its features, then predict the uplifts of each treatment through a single head. This paper considers the multi-treatment multi-task uplift modeling and explores its application in our online gaming server.

## 2.2 Multi-task Learning

Multi-task learning has been widely applied in computer vision, recommendation systems, and other fields. To boost the online business, multiple targets often influence marketing strategies. While focusing on a single task, existing models tend to ignore useful information from the training signals of related tasks Ruder (2017). The introduction of multiple tasks mitigates the sample bias, where tasks with more training samples and be informative to tasks with fewer samples Zhang et al. (2023). In addition, training a multi-task model can reduce the cost of maintaining several models for each task. In the era of deep learning, a popular research line of multi-task learning is parameter sharing, where parameters are shared between different tasks. This includes hard parameter sharing Guo et al. (2020), which encodes representations of tasks into a shared embedding, and then applies task-specific heads to predict the outcome of each task. Extended from this, soft parameter sharing Duong et al. (2015) applies separate branches to model tasks and share information between branches by weighted addition or attention. Additionally, there is expert sharing Ma et al. (2018), which utilizes several expert models to embed features and weigh the influence of each sub-task, and subsequently apply weights to calibrate the output of each expert.

While proven effective in other domains, research on multi-task uplift modeling is still limited. We adopt the multi-gate mixture-of-expert in our framework to handle multiple tasks and individually estimate uplifts on each task.

# 3 Problem Definition

Assume the observed data to be $\mathcal{D} = \{[x_i, (\hat{t}_i, t_i), y_i]\}_{i=1}^N$, where $x_i \in \mathbb{R}^d$ is the $d$-dimensional user features, $\hat{t}_i \in \{0, 1\}$ is the base treatment that denotes whether offering incentives to users, $t_i \in \mathbb{N}^m$ is the secondary treatment features with $m$ discrete treatments, $y_i \in \mathbb{R}^k$ is the $k$ tasks, and $N$ is the number of samples. Treatments are exclusive and each individual can receive only one treatment at a time.

The basic assumptions of causal inference are satisfied to ensure the identifiability of base ITE and incremental ITE estimation Wang et al. (2024); Shalit et al. (2017). Following the Neyman-Rubin causal inference framework Rubin (2005), we define the potential outcomes. To properly decompose the overall treatment effect, we conceptualize it in two stages. First, we model the effect of receiving any treatment versus no treatment (control). Second, we model the additional effect of one specific treatment over another.

let $y_i^k(0)$ denotes the potential outcome of the $i$-th user in the control group on task $k$, and $y_i^k(m)$ denotes the outcome of receiving treatment $m$. To bridge these two stages, we introduce $y_i^k(1)$ as the potential outcome under a general treatment condition, which serves as a common reference point for the incremental effects of all specific treatments $m$. Due to the counterfactual problem, we can only observe one of these potential outcomes for each user. Instead of training on ground truth, we estimate the expected response differences.

Let $\hat{\tau}^k(x_i)$ be $i$-th user's uplift when receiving a treatment, $\tau_m^k(x_i)$ be the incremental uplift under the $m$-th treatment and $k$-th task. Then the overall treatment effect can be computed as:

$$\begin{aligned}
\Gamma_m^k(x_i) &= \hat{\tau}^k(x_i) + \tau_m^k(x_i) \cdot \mathbb{I}(\hat{t} = 1) \\
&= \mathbb{E}(y_i^k(1) - y_i^k(0)|x_i) + \mathbb{E}(y_i^k(m) - y_i^k(1)|x_i, \hat{t} = 1)
\end{aligned} \tag{1}$$

where $\mathbb{I}$ is the indicator function, $\Gamma_m^k(x_i)$ is the $i$-th user's overall uplift score on the $k$-th task, $\hat{t}$ signifies if an individual is in the treatment group. The base uplift $\mathbb{E}(y_i^k(1) - y_i^k(0))$ captures the effect of moving a user from the control group to the general treatment group. The incremental uplift $\mathbb{E}(y_i^k(m) - y_i^k(1))$ captures the marginal effect of the specific treatment $m$ relative to the common base treatment effect.

## 4 Methodology

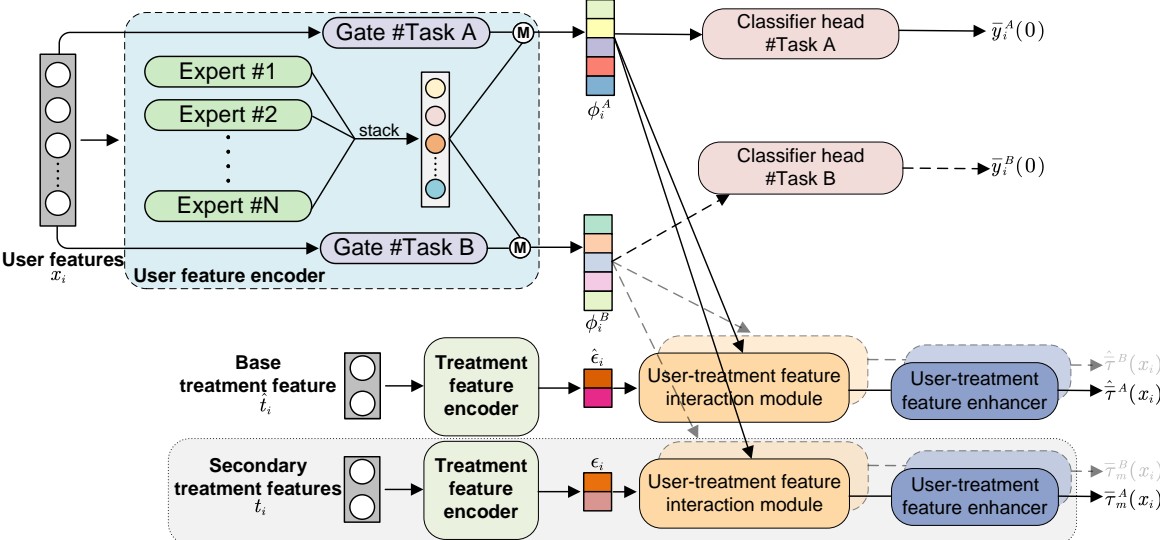

Figure 2: Illustration of the proposed **M**ulti-**t**reatment **M**ulti-**t**ask (MTMT) framework. Note that for clarity we only show the network structure of two tasks.

### 4.1 Architecture

The illustration of the proposed multi-treatment multi-task (MTMT) uplift modeling network is shown in Fig. 2. Given a sample $\{x_i, (\hat{t}_i, t_i), y_i\}$, the user features $x_i$ are first encoded by the user feature encoder to generate the representations for each task $\{\phi^0, \phi^1, \cdots, \phi^k\}$. Meanwhile, the base treatment feature $\hat{t}_i$ (indicates control or treatment) and the secondary treatment features $t_i$ (indicates specific types of treatments) are encoded separately to generate the corresponding representations $\hat{\epsilon}$ and $\epsilon$. The feature representations of each task are projected by the corresponding classifier head to compute its natural response $\bar{y}_i^k(0)$ of task $k$ when not treated. Additionally, the embeddings of base and secondary treatments are fed separately into the corresponding user-treatment interaction module, in which the cross-correlations between treatment and user features are computed. The resultant treatment-aware features are further enhanced and used to estimate the response difference for receiving treatment and the incremental uplift for receiving treatment $m$.

### 4.2 Task-Oriented Feature Encoder

To explicitly model inter-task relationships and learn task-specific representations, we adopt the multi-gate mixture-of-experts (MMOE) Ma et al. (2018) as the user feature encoder. Mixture-of-experts is a form of ensemble learning that integrates numerous expert models (e.g., vanilla CNN) to learn a shared representation

and use the combined predictions to improve accuracy Eigen et al. (2013). Extended from this, MMOE introduces an additional gating network to filter useful information from the shared representation of each task. To obtain the encoded representation $\phi_i^k$ for task $k$:

$$\phi_i^k = \mathcal{G}^k(x_i) \cdot \{\mathcal{E}^1(x_i), \mathcal{E}^2(x_i), \cdots, \mathcal{E}^E(x_i)\} \tag{2}$$

where $\mathcal{E}^j$ is the $j$-th expert network and there are $E$ experts and $\mathcal{G}^k$ is the gating network for $k$-th task. We empirically choose ResNet18 He et al. (2016) as the backbone for each expert. Note that the embeddings from the experts are stacked and then filtered by the corresponding gate to produce a task's representation. The gating network can be a simple linear projection from the input with additional activation:

$$\mathcal{G}^k = \text{softmax}(W_g^k x_i) \tag{3}$$

where $W_g^k \in \mathbb{R}^{n \times d}$ is the trainable weights, $n$ is the number of experts, and $d$ is the feature dimension.

The representations $\phi_i^k$ for each task only contain non-treatment information and are therefore used to estimate the natural response of the control groups by a linear projection head:

$$\bar{y}_i^k(0) = W_i^{\text{proj}} \phi_i^k \tag{4}$$

Since most treatments are binary or discrete, we first one-hot encode the base and secondary treatment separately and multiply the resultant sparse vector with a corresponding learnable dense matrix $\hat{A}_i \in \mathbb{R}^{v \times 1}, A_i \in \mathbb{R}^{v \times m}$ to produce embeddings:

$$\hat{\epsilon}_i = \hat{A}_i \hat{t}_i, \quad \epsilon_i = A_i t_i \tag{5}$$

where $v$ is the embedding dimension and $m$ is the number of possible treatments.

It should be noted that both user and treatment feature encoders can be substituted with other feature representation learning networks, provided they have the appropriate dimensions. For instance, a single ResNet18 can be used for user feature extraction, adapting the Multi-Task Multi-Treatment (MTMT) model to a single-task setting. The flexibility of the proposed design allows for seamless online implementation across various problem settings.

### 4.3 User-Treatment Feature Interaction

To explicitly utilize treatment features and model their relationships with user features, we propose the user-treatment feature interaction module based on co-attention Rombach et al. (2022). The structure is shown in Fig. 3. The decision to use the treatment embedding $\epsilon_i$ as the query and the user feature $\phi_i^k$ as the key and value is a deliberate design choice to enhance model interpretability. In this configuration, the attention mechanism aims to answer the question: "Given this specific treatment, which of the user's characteristics are most important for determining its effect?"

We treat the treatment embeddings $\epsilon_i$ as the query and the user feature embeddings $\phi_i^k$ as the key and value. Subsequently, we compute how each treatment attends to each user feature, then use the resultant attention scores to generate treatment-aware embeddings:

$$\psi_{i,m}^k = \text{softmax}\left(\frac{W_i^{\mathcal{T}} \epsilon_i \times (W_i^{\mathcal{U}} \phi_i^k)^T}{\sqrt{d_{\mathcal{U}}}}\right) W_i^{\mathcal{U}'} \phi_i^k \tag{6}$$

where $W_i^{\mathcal{T}}$ linearly projects the treatment embedding and $W_i^{\mathcal{U}}$, $W_i^{\mathcal{U}'}$ linearly projects the user feature embeddings. $\sqrt{d_{\mathcal{U}}}$ is a scaling factor. The output $\psi_{i,m}^k$ is a new representation of the user, re-weighted according to the specific treatment being considered. This highlights the user characteristics most relevant for that treatment.

We then process treatment-aware features $\psi_i^k$ through two separate enhancement and projection paths to isolate the base and incremental effects. Specifically, two distinct Multi-Layer Perceptrons (MLPs) act as

feature enhancers, one for the base uplift and one for the incremental uplift. The resulting refined embeddings are projected to estimate the base uplift score $\hat{\bar{\tau}}^k(x_i)$ and the incremental uplift score $\bar{\tau}_m^k(x_i)$ for treatment $m$:

$$\hat{\bar{\tau}}^k(x_i) = \hat{W}_i^{\mathcal{T}} \mathrm{MLP}_{\mathrm{base}}(\bar{\psi}_{i,m}^k), \quad \bar{\tau}_m^k(x_i) = W_i^{\mathcal{T}} \mathrm{MLP}_{\mathrm{incr}}(\psi_{i,m}^k) \tag{7}$$

where $W_i^{\mathcal{T}}$ and $\hat{W}_i^{\mathcal{T}}$ are the projection matrix.

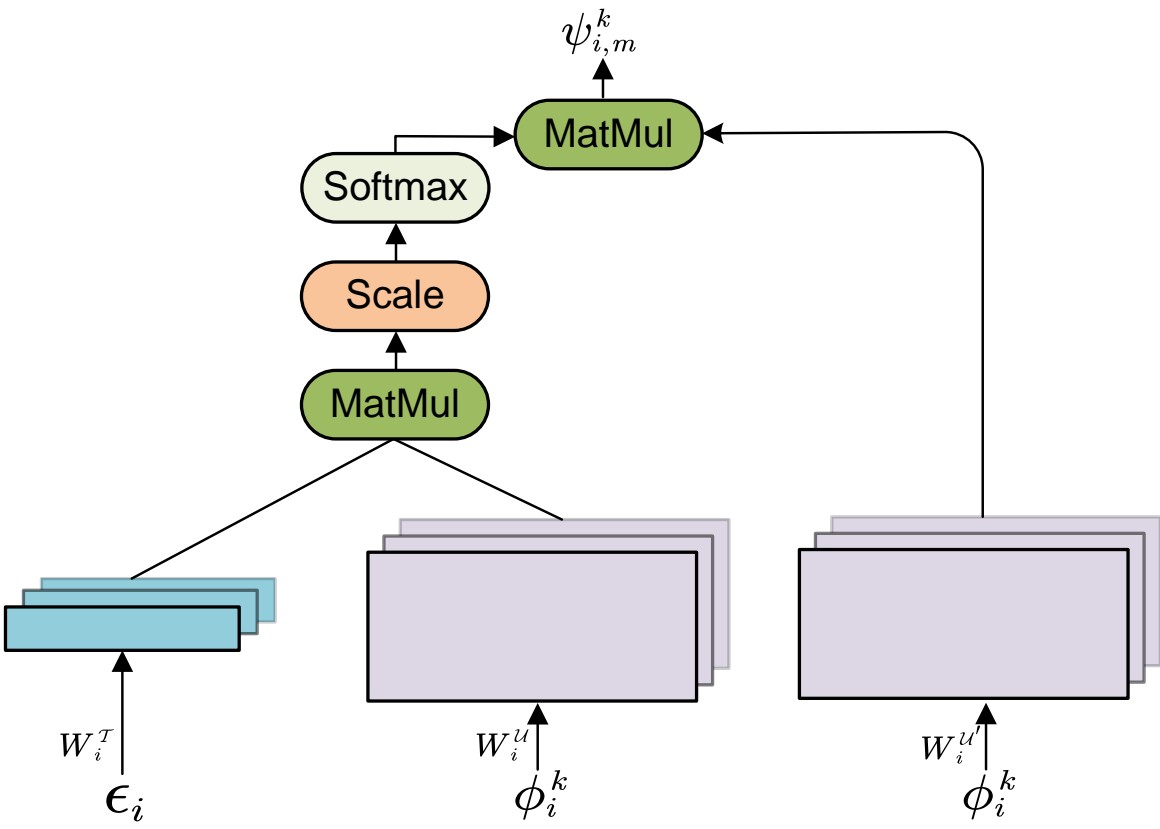

Figure 3: The proposed user-treatment feature interaction module

## 4.4 Multi-Treatment Multi-Task Uplift Estimation

Given the natural response $\bar{y}_i^k(0)$, the base uplift score $\hat{\bar{\tau}}^k(x_i)$, and the incremental uplift score $\bar{\tau}_m^k(x_i)$, we can estimate the user's response by:

$$\bar{y}_i^k(m) = \bar{y}_i^k(0) + (\hat{\bar{\tau}}^k(x_i) + \bar{\tau}_m^k(x_i)) \cdot \mathbb{I}(\hat{t} = 1) \tag{8}$$

Here $\hat{\bar{\tau}}^k(x_i)$ indicates how the user acts to a treatment, and $\bar{\tau}_m^k(x_i)$ indicates the incremental effect of the specific treatment $m$, on top of the base uplift. Intuitively, $\bar{\tau}_m^k(x_i)$ is only effective when the user is in the treatment group. For online deployment when the treatment information is unknown, we permute all possible combinations of treatments and rank the resultant base and incremental uplifts.

Compared to previous methods for multi-treatment uplift modeling, the proposed approach separately estimates the effects of "whether to treat" and "what to treat" using the base and secondary treatment features, thereby bridging the numerical gap between the base uplift and the uplift variations among treatments.

### 4.5 Training and Inference

We use the natural response and treated response for each task to compute the overall loss across the entire sample space:

$$
\sum_{k=1}^{N} [\sum_{i \in \mathrm{C}} \mathcal{L}(y_i, \bar{y}_i^k(0)) + \sum_{i \in \mathrm{T}} \mathcal{L}(y_i, \bar{y}_i^k(m))]
$$
$$
= \sum_{k=1}^{N} [\sum_{i \in \mathrm{C}} \mathcal{L}(y_i, \bar{y}_i^k(0)) + \sum_{i \in \mathrm{T}} \mathcal{L}(y_i, \bar{y}_i^k(0) + \hat{\bar{\tau}}^k(x_i) + \bar{\tau}_m^k(x_i))]
$$
(9)

where $\mathcal{L}$ is the mean-squared error loss function, C denotes the control group, and T denotes the treatment group. By optimizing this objective, the network learns to decompose the observed outcome into the base non-treated response, the effect of the base treatment, and the incremental effect of the secondary treatment, thereby learning to estimate the causal quantities of interest. At inference, we only use Eq.7 to directly compute the base and incremental uplifts, which are subsequently ranked and used to determine whether to offer treatment and which treatment to offer.

## 5 Experiments

We conduct extensive experiments to verify the effectiveness of the proposed MTMT in single-treatment single-task, single-treatment multi-task, multi-treatment single-task, and multi-treatment multi-task settings. We mainly focus on the following questions:

- **RQ1:** Can the proposed MTMT outperform other baseline methods on public and product datasets?

- **RQ2:** How does each design contribute to the overall performance of MTMT?

- **RQ3:** Whether the model produces interpretable results that are consistent with our online observations?

### 5.1 Experimental Setup

#### 5.1.1 Datasets

- **CRITEO** Diemert et al. (2021): CRITEO is an open-sourced uplift modeling dataset for online advertising. The data is created by compiling data from various incremental tests, with a specific type of randomized trial in which a portion of the population is randomly excluded from advertising targeting. We include about 14 million samples, each has 12 continuous features, and use visit as the target.

- **Product**: We include a product dataset containing over 10 million samples randomly collected from the our online gaming platform. There are about 700 discrete and continuous features that describe users' static profiles and their recent gaming histories. To minimize the influence of confounding factors in uplift modeling, we gather data from randomized controlled trials (RCTs). In these trials, treatments are assigned randomly and each user has the non-zero probability to receive each treatment. For the multi-task scenario, we employ two binary labels: whether the user logs in the next day (short-term activity) after receiving or not receiving the treatment and whether the user plays more games in the next 7 days (long-term activity). For the multi-treatment scenario, we identify the base treatment as "whether to give a bonus" and the secondary treatment as "the type of bonus" (bonus type A and bonus type B), both of which are binary-valued. Note that in the practical application, certain bonus types are only available for a subset of users. Therefore, we collect an additional multi-treatment dataset by selecting users who are accessible by all types of bonuses.

Table 1: Overall performances of the single-treatment single-task version of MTMT and its comparatives on the public and product datasets. We use two targets of the product dataset, with a single treatment (whether to offer a bonus). Note that the last row is the multi-task version of MTMT. The best baselines are tilted and the best methods are marked as bold for each metric.

| Dataset | CRITEO | | | Product - short-term activity | | | Product - long-term activity | | |
|---|---|---|---|---|---|---|---|---|---|
| Metrics | QINI | AUUC | LIFT@30 | QINI | AUUC | LIFT@30 | QINI | AUUC | LIFT@30 |
| S-Learner | 0.0703 | 0.0283 | 0.0258 | 0.0420 | 0.0880 | 0.00575 | *0.131* | 0.0101 | 0.0873 |
| T-Learner | 0.0706 | 0.0286 | 0.0271 | 0.0421 | *0.0890* | 0.00763 | 0.111 | 0.00837 | 0.0831 |
| CFR | 0.0715 | 0.0295 | 0.0278 | 0.0100 | 0.0182 | 0.0037 | 0.108 | *0.0211* | *0.0915* |
| DragonNet | 0.0183 | 0.00735 | 0.0121 | 0.00743 | 0.0156 | 0.00393 | 0.0812 | 0.0164 | 0.0792 |
| EUEN | 0.0730 | 0.0297 | 0.0279 | 0.00827 | 0.0262 | 0.00313 | 0.106 | 0.0113 | 0.0778 |
| DESCN | 0.0718 | 0.0289 | 0.0264 | 0.0351 | 0.0771 | 0.00438 | 0.0073 | 0.0110 | 0.0177 |
| FlexTENet | *0.0779* | *0.0322* | *0.0290* | 0.0321 | 0.0684 | 0.00268 | 0.111 | 0.0106 | 0.0796 |
| EFIN | 0.0725 | 0.0293 | 0.0215 | *0.0725* | 0.0293 | *0.0415* | 0.0340 | 0.0281 | 0.0554 |
| M3TN | 0.0395 | 0.0176 | 0.0205 | 0.0295 | 0.0416 | 0.00260 | 0.108 | 0.0204 | 0.0908 |
| **MTMT (single treatment)** | **0.164** | **0.0593** | **0.0338** | **0.0886** | **0.155** | **0.0638** | **0.360** | **0.0579** | **0.110** |
| **MTMT (multi-task)** | – | – | – | **0.0586** | **0.118** | **0.0326** | **0.154** | **0.0218** | **0.111** |

### 5.1.2 Baselines

To demonstrate the performance of MTMT, we include a set of popular methods proposed for uplift modeling, including: S-Learner Künzel et al. (2019), T-Learner Künzel et al. (2019), CFR Shalit et al. (2017), DragonNet Shi et al. (2019), EUEN Ke et al. (2021), DESCN Zhong et al. (2022), FlexTENet Curth & Van der Schaar (2021a), EFIN Liu et al. (2023), and M3TN Sun & Chen (2024).

For the multi-treatment problem, we employ EFIN, M3TN, and HydraNet Velasco-Regulez & Cerquides (2023). Furthermore, we extend S-Learner and T-Learner to handle multiple treatments. For S-Learner, we directly apply the multi-treatment features as the input and iterate all possible treatment assignments to compute the corresponding uplift score. For T-Learner, we use multiple branches to process each treatment individually.

### 5.1.3 Evaluation Metrics

Following the previous works, we adopt Area Under the QINI Curve (QINI), Area Under the Uplift Curve (AUUC), and the uplift score at first 30% (LIFT@30) to evaluate the uplift ranking capability of different models. Note that for easier and fairer comparison, we employ the normalized QINI and AUUC.

### 5.1.4 Implementation Details

We train all models on NVIDIA A100, with Pytorch 2.1.2 and Python 3.11. We use the AdamW optimizer Kingma & Ba (2014) with a learning rate of 0.001 and cosine annealing learning rate scheduler Loshchilov & Hutter (2016). Additionally, we use a batch size of 15360 and set the maximum epochs as 50. For the detailed parameter setting of MTMT, we employ the standard ResNet without the classifier head as the expert in the user feature encoder and set the number of experts to 4.

### 5.2 RQ1: Performance Comparison

We evaluate the single-treatment single-task variation of the proposed methods and the baselines for single-treatment and multi-treatment uplift estimation. The results are presented in Table 1 and Table 2. Note that for MTMT in the single-treatment case, we only use base uplift $\hat{\bar{\tau}}^k(x_i)$ from Eq.7 to decide whether a treatment should be offered.

In Table 1, we test all models based on two tasks on the product dataset: short-term activity and long-term activity. We further show how the multi-task version of MTMT performs on the two tasks. From the table, MTMT has 0.164 QINI on the CRITEO dataset. Meanwhile, the best-performing baseline on the CRITEO dataset is FlexTENet, which reaches 0.0779 QINI. On the product dataset, MTMT still maintains

Table 2: Overall performances of MTMT and its comparatives on the multi-treatment product datasets. The best baselines are tilted and the best methods are marked as bold for each metric.

| Treatment | Bonus A | | | Bonus B | | |
|---|---|---|---|---|---|---|
| Metrics | QINI | AUUC | UPLIFT@30 | QINI | AUUC | UPLIFT@30 |
| S-Learner | 0.0126 | 0.0236 | 0.00211 | 0.00771 | 0.0152 | 0.00105 |
| T-Learner | 0.00505 | 0.00878 | 0.00716 | 0.00374 | 0.00942 | 0.00141 |
| HydraNet | 0.00456 | 0.00844 | 0.0182 | 0.00616 | 0.012 | 0.00895 |
| EFIN | 0.0142 | 0.0256 | *0.00766* | *0.0294* | *0.0513* | *0.0334* |
| M3TN | *0.0162* | *0.0281* | 0.00259 | 0.00107 | 0.00640 | 0.0175 |
| **MTMT (multi-treatment)** | **0.0324** | **0.0653** | **0.0344** | **0.0291** | **0.0782** | **0.0491** |

its advantage over other baselines on both tasks. For the multi-task setting, MTMT's performance slightly degrades, while its overall performance is still much better than the baselines.

To further validate MTMT's performance in the multi-treatment setting, we present the results in Table 2. We separately calculate the metrics on the two types of treatments, bonus A and bonus B. From the table, among the baselines, M3TN performs the best on bonus A and EFIN performs the best on bonus B, while MTMT outperforms all its comparatives on the two types of treatments.

### 5.3 RQ2: Ablation Study

We conduct ablation studies to validate the effectiveness of the key design of MTMT. Specifically, for the architecture, we remove the user-treatment feature enhancer and substitute the user-treatment feature interaction module with matrix multiplication. We also test using MLP (which is more widely used in prior research Liu et al. (2023)) as the feature encoder. For the multi-treatment case, we remove the "secondary treatment features" branch as in Fig.2 and estimate uplifts of multiple treatments with a single output. For the multi-task case, instead of estimating ITE for each task, we jointly estimate the ITE for all tasks. The results are presented in Table3. From the table, changing the network architecture or modifying the ITE estimation process can result in performance degradation.

Table 3: Ablation study of MTMT on the product dataset. For the multi-treatment setting, we show the model's performance on different treatments as "bonus A/bonus B". For the multi-task setting, we show the model's performance on different tasks as "short-term activity/long-term activity". The proposed designs are marked in bold.

| | | Product | | |
|---|---|---|---|---|
| | | QINI | AUUC | LIFT@30 |
| architecture | w/o user-treatment feature interaction | 0.0426 | 0.0881 | 0.00849 |
| | w/o user-treatment feature enhancer | 0.0759 | 0.145 | 0.0474 |
| | MLP as feature encoder | 0.0819 | 0.152 | 0.0627 |
| | **MTMT** | **0.0886** | **0.155** | **0.0638** |
| multi-treatment | w/o tiered treatment effect estimation | 0.0249/0.0233 | 0.0464/0.0514 | 0.0164/0.0186 |
| | **MTMT (multi-treatment)** | **0.0324/0.0291** | **0.0653/0.0782** | **0.0344/0.0491** |
| multi-task | w/o task-wise treatment effect estimation | 0.0552/0.108 | 0.113/0.0118 | **0.035/0.0922** |
| | **MTMT (multi-task)** | **0.0586/0.154** | **0.118/0.0218** | 0.0326/0.111 |

### 5.4 RQ3: Interpretable Analysis

To demonstrate the model produces interpretable results that are consistent with our online observations, we visualize the density distributions of the base treatment effect ($\hat{\bar{\tau}}^k(x_i)$) and the incremental treatment effect ($\bar{\tau}_m^k(x_i)$) using our product dataset. As shown in Fig. 4, the average of $\hat{\bar{\tau}}^k(x_i)$ (0.055) is much larger than the average of $\bar{\tau}_m^k(x_i)$, which corresponds to our observations in the online gaming server (Fig. 1).

To validate the intended behavior of the user-treatment feature interaction module, we conduct an interpretability analysis by visualizing the attention scores from Eq. 6. This analysis examines which user

characteristics the model deems most salient when estimating the effects of different treatments for various tasks. As illustrated in Fig. 5, the module learns to assign non-uniform attention weights across the user feature space. We observe distinct attention patterns that vary according to both the specific treatment and the prediction task. For instance, the subset of user features receiving high attention for 'Bonus A' differs from that for 'Bonus B'. This suggests that the module effectively identifies and leverages different user characteristics to model the impact of each unique treatment.

Furthermore, the attention distributions are task-dependent. When considering the same treatment, the model prioritizes different user features for each of the two tasks. This confirms that the module is generating representations that are not only treatment-aware but also tailored to the specific objective of each task. These findings support the conclusion that the user-treatment interaction mechanism is successfully creating contextualized user representations ($\psi_{i,m}^k$), which is a critical step for accurately decomposing the base and incremental uplift effects within the proposed MTMT framework.

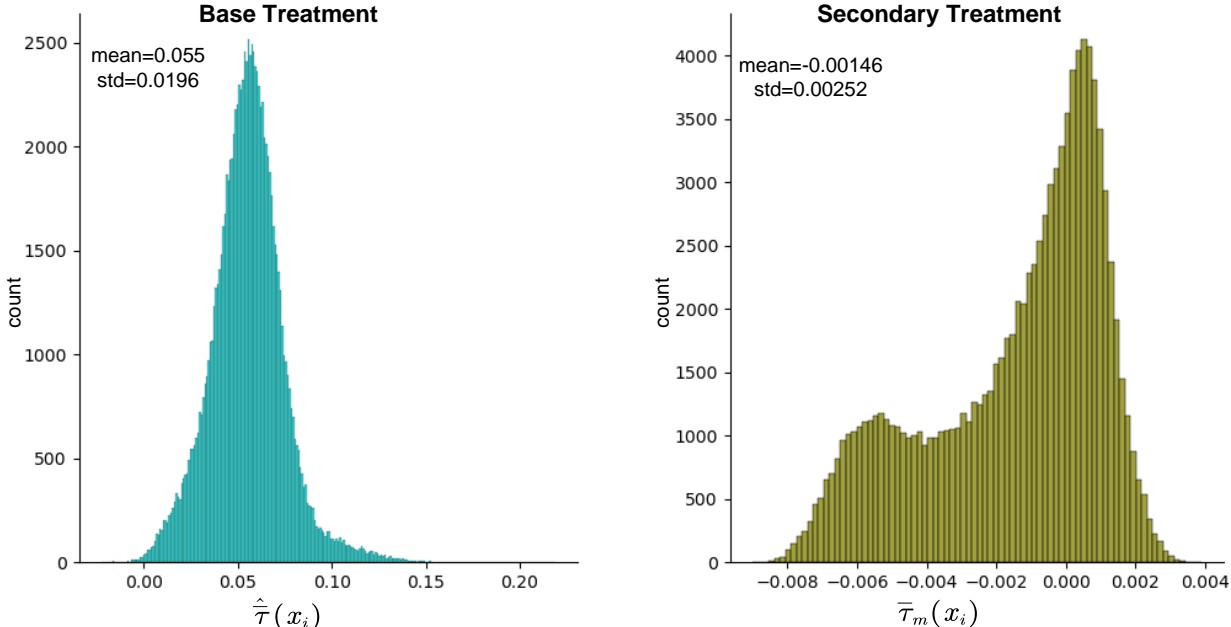

Figure 4: Distributions of base and incremental treatment effects. The base treatment effects are numerically much larger than the incremental treatment effects.

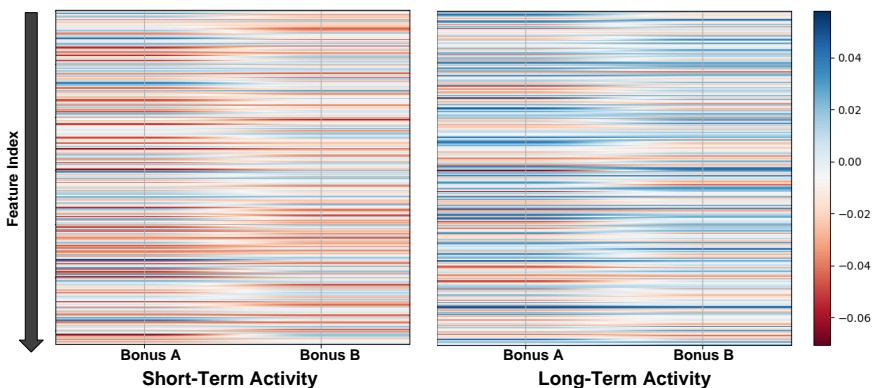

Figure 5: Attention score of user-treatment feature interaction module under the multi-treatment multi-task setting.

### 5.5 Online Deployment

The primary objective of our online deployment is to leverage uplift modeling to personalize bonus allocation, thereby enhancing the user's flow experience. To achieve this, we offer various bonuses designed to improve the overall gaming experience. The process for this online deployment is depicted in Figure 6 . Initially, the Multi-Task Multi-Treatment (MTMT) model is trained offline using historical data on user features and treatment effects. Following this training, the model segments users into distinct buckets based on their estimated baseline behavior and predicted response to different treatments. For each user bucket, the gaming server then determines the optimal strategy, which includes deciding whether to issue a bonus and, if so, which specific bonus to provide, all while adhering to predefined operational constraints.

To validate our approach in a live environment, we conducted an online A/B experiment. We established four independent sets of user buckets within our production environment. Each bucket contained millions of users and maintained a consistent distribution of user features. In this experiment, we compared the performance of four distinct policies: a random policy (control group), an EFIN policy, an MTMT single-task policy, and our proposed MTMT multi-task policy.

The experiment involved two different bonuses. We evaluated the effectiveness of each policy based on two key metrics: short-term user activity, measured by the percentage of users who logged in the following day, and long-term user activity, measured by the number of monthly active users. The results, presented in the table below, demonstrate statistically significant improvements for our proposed models (p < 0.01).

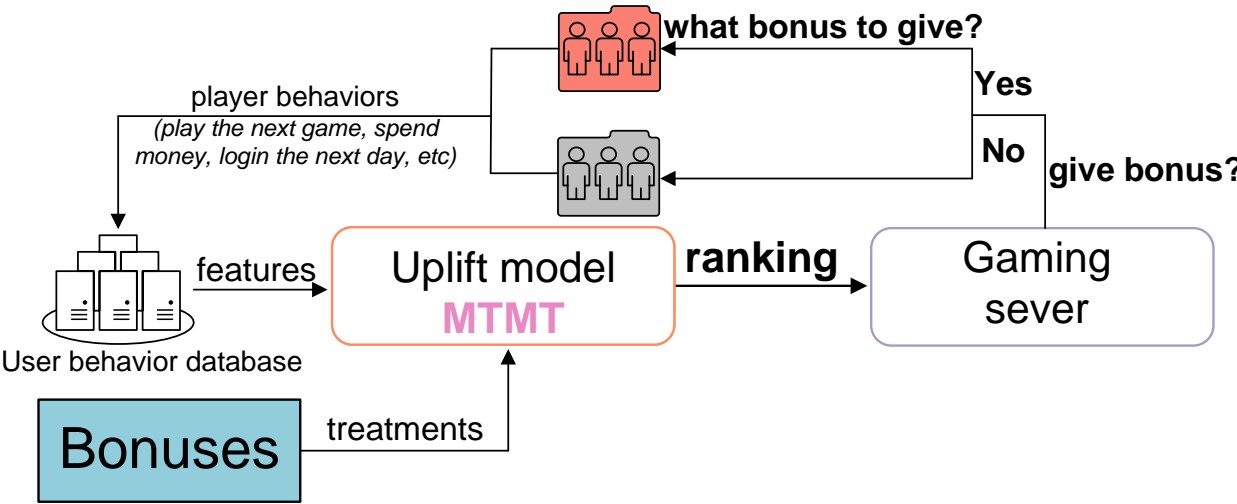

Figure 6: Overview of our online bonus deployment platform

Table 4: Results of MTMT in an online deployment

| Policy | Avg. Short-Term Activity | Long-Term Activity |
|---|---|---|
| Random | 0.0% | 0.0% |
| EFIN | +8.7% | +6.4% |
| MTMT (single-task) | +15.5% | +10.5% |
| MTMT (multi-task) | +16.1% | +11.9% |

## 6 Conclusion and Future Work

In this paper, we explored uplift modeling within a multi-treatment, multi-task framework. To effectively extract features for different tasks and accurately estimate the effects of various treatments, we proposed the Multi-Treatment Multi-Task (MTMT) uplift modeling framework that explicitly utilizes user-treatment

features. Additionally, to precisely capture the uplift differences between various treatments, we proposed separately estimating the base treatment effect and the incremental treatment effect. Extensive experiments and ablation studies validated the effectiveness of our methods. In future work, we plan to extend our models to accommodate non-binary treatments and non-binary tasks, thereby increasing their applicability to a broader range of scenarios.

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
