# OpenReview forum: "MTMT: Tiered Treatment Effect Decomposition for Multi-Task Uplift Modeling"
_TMLR — Withdrawn by Authors_

### Review · Reviewer_swJz · 2025-08-16

**Summary Of Contributions:**

This paper proposes a novel framework for uplift modeling in scenarios with multiple treatments and multiple outcome tasks. The authors introduce a Multi-Treatment Multi-Task (MTMT) uplift modeling network that decomposes the treatment effect into a tiered structure: a base effect (the effect of receiving any treatment vs. none) and an incremental effect for each specific treatment type. The model’s architecture uses a multi-gate mixture-of-experts (MMoE) network to learn shared and task-specific user feature representations for each prediction task.

**Audience:**

Yes

**Audience Explanation:**

This paper’s findings should interest a segment of TMLR’s audience, particularly those working in machine learning for causal inference, recommender systems, or marketing analytics. While the topic is somewhat niche (uplift modeling for personalized treatments), it addresses a very practical problem faced in industry, and it introduces a novel approach that crosses between causal modeling and multi-task deep learning.

**Claims And Evidence:**

Yes

**Claims Explanation:**

The authors back up their main technical claims with extensive experimental results and ablation studies. The performance improvements of MTMT over baseline methods are clearly shown with appropriate evaluation metrics, and the paper includes analyses (like ablations and interpretability visualizations) that support the authors’ explanations for why their model works. A few secondary claims (such as real-world deployment success) are asserted with less detail, but overall the evidence provided is convincing and aligns with the paper’s core claims.

**Requested Changes:**

1/ The authors should clarify how the baseline models were implemented and tuned. Given the large performance gap, it’s important to assure readers that each baseline received adequate hyperparameter tuning and that any architectural advantages (e.g. larger model capacity in MTMT) are acknowledged.

2/ Since the multi-task version showed slightly lower performance on each individual task than dedicated single-task models, it might be helpful to briefly discuss why. For instance, do the authors suspect it’s due to negative transfer, or simply because multi-task MTMT is jointly optimizing more objectives.

3/ To facilitate adoption of the MTMT model by other researchers, the authors could consider providing pseudocode in an appendix or releasing their implementation as open-source.

---

### Review · Reviewer_i8BC · 2025-08-22

**Summary Of Contributions:**

This paper identifies several limitations in prior uplift modeling research:

- limited problem formulation and scope. Most existing research focuses on a single-treatment, single-task setting, which does not reflect the complexity of real-world marketing where multiple incentives and business goals are involved.
- the effect of receiving *any* incentive versus no incentive (the "base effect") is often numerically much larger than the difference in effect *between* different incentives (the "incremental effect"). Lumping these two effects together makes it difficult for a model to learn the subtle but important differences between treatments, leading to suboptimal estimations.
- existing research overlooks the implicit relationship between treatments and various tasks and fails to capture the nuanced interactions required for accurate estimation.

To address these issues, this paper proposes to:

- a. use MMOE network as a task oriented user feature encoder,
- b. two-tiered separate treatment encoders that explicitly encodes the base treatment and secondary treatment
- c. user-treatment interaction module that uses self attention to create treatment-aware user representation.

The final prediction is thus the sum of natural response, base uplift and the relevant incremental uplift.

The paper is evaluated on two datasets: CRITEO (public single task, single treatment dataset),  Product (propitiatory multi-task, multi-treatment dataset). Online experiment, ablation study are also reported.

## Strengths

- Uplift modeling in online marketing using neural models is a very relevant and trending research topic.
- The technical construction of the proposed approach is sensible.
- The empirical study shows the potential effectiveness of the proposed approach

## Weaknesses

- No guarantee on sample imbalance or unbiased estimation discussed in the paper, despite they are claimed issues of the existing approach.
- Unclear or overly strong claims about the limitation of the existing work as well as about the contribution of the present work.
- Experiments do not fully support the main claims in the proposed approach.

**Audience:**

Yes

**Audience Explanation:**

Despite the issues above, given the reported empirical study (valid on propitiatory data) as well as the interesting architecture,  in my opinion, at least researchers and practitioners in uplift modeling in online marketing as well as uplift modeling / causal inference in general would find this paper interesting.

**Broader Impact Concerns:**

No concerns.

**Claims And Evidence:**

No

**Claims Explanation:**

- The paper positions itself as a solution for the multi-treatment, multi-task setting, but the only public, verifiable benchmark they use (CRITEO) does not test this core capability. The authors test a simplified "single treatment" version of their model on CRITEO. While its strong performance there is a good sign, it does not validate the central claims about handling multiple treatments and tasks.
- Given the previous point, a synthetic data experiment is highly needed. As ground truth counterfactuals are not available in the real world dataset, controlled synthetic data experiments is a standard and highly effective way to validate the proposed approach. see, for example:
    - Exploring Transformer Backbones for Heterogeneous Treatment Effect Estimation
    - Meta-learners for Estimating Heterogeneous Treatment Effects using Machine Learning
- There is no guarantee nor analysis about the performance of the proposed approach under sample imbalance between the treatment and control groups, as well as the unbiased estimation of ITE, which are the claimed limitation of existing approaches in the paper.
- In abstract, the claim "However, previous research typically considers a single-task, single-treatment setting..." is too strong, which contradicts with the narrative in the introduction, where there are plenty of works on multi task and multi treatment problems.
- In the contribution statements in introduction, claim "To the best of our knowledge, this is the first effort in multi-treatment multi-task uplift modeling without underlying assumptions about the treatments or tasks." is unclear. What are the "underlying assumptions" that is dropped by the proposed approach? If so, at what cost?

**Requested Changes:**

Please address the rationales provided in the “Claim soundness” assessment section.

---

### Review · Reviewer_ZYt3 · 2025-08-24

**Summary Of Contributions:**

This work develops a method for estimating the individualized treatment effect (ITE) in randomized experiments, extending existing literature to accommodate settings with multiple treatments and tasks. The authors decompose the multi-treatment into two parts: a base effect representing the impact of offering any treatment and an incremental effect capturing the additional impact of assigning a specific treatment type. To account for heterogeneity in treatment response, the method further incorporates a user–treatment interaction module that models treatment response variation as a function of user features. Empirical evaluations demonstrate improved performance relative to existing approaches developed for either multi-treatment or multi-task settings alone.

**Audience:**

Yes

**Audience Explanation:**

This work addresses the gap in individualized treatment effect estimation under multi-treatment and multi-task settings by introducing the MTMT method, which shows good empirical performance.

**Claims And Evidence:**

No

**Claims Explanation:**

**Weaknesses:**
1. The definitions and identification assumptions in the Problem Definition section could be made more explicit. In particular, it would be helpful to formally define the individualized treatment effect (ITE) using potential outcome notation, e.g., $Y_i(a) - Y_i(a')$ for individual $i$, comparing treatment $a$ versus $a'$. Since the ITE is not identifiable from the observed data, the authors should clarify that what is actually being estimated is the conditional average treatment effect (CATE), defined as $\E(Y(a) - Y(a') \mid X_i = x)$.
    For instance, when the paper states that “low-active users are more likely to play the next game when given a higher-valued bonus, whereas high-active users may continue to play regardless of receiving a bonus,” this is describing heterogeneity in treatment response across users with different features, which corresponds to the CATE rather than the ITE. Making this distinction explicit, and presenting the estimand clearly in the potential outcome notation, would strengthen the clarity and rigor of the work.

2. The base effect and incremental effect are key innovations of this work, but their definitions are not explicitly given. The authors describe these functions informally as $\bar{\tau}^k(x_i)$ being the $i$-th user’s uplift when receiving a treatment, and $\bar{\tau}_m^k(x_i)$ being the incremental uplift under the $m$-th treatment and $k$-th task. However, the explicit mathematical definitions are not clear from Equation~(1). Also, how the $\mathbb{I}(\hat{t}=1)$ being moved to the condition set is not clear. It would strengthen the presentation to provide formal definitions in the potential outcome notation.

3. The definitions in Equation (1) and (8) appear inconsistent. Based on Equation(1), the $\bar{\tau}$ terms seem to represent _average_ counterfactual outcomes, i.e. quantities  defined in $\mathbb{E}()$. In contrast, Equation~(8) suggests that they are defined directly as counterfactual outcomes. Again, providing explicit and consistent definitions is important.

4. I am curious about the intuition behind Equation (6). It appears to represent the interaction between treatment and user features, but the rationale for adopting this specific functional form is unclear. Why is this formulation chosen over a simpler alternative, such as a direct multiplicative interaction between the treatment and user feature representations?

5. Why is it necessary to construct a treatment embedding in Equation (5), given that the treatment variable in the application is discrete with only two possible values (bonus A/B)? In such a case, a direct representation might seem sufficient. Clarifying the benefit of using an embedding would be helpful.

6. I wonder how the proposed MTMT method compares to an approach that treats each treatment level separately against the control and applies existing single-treatment and multi-task uplift methods, given that the treatment considered in current work is discrete. For example, one could estimate single-treatment multi-task CATEs by modeling the effects of bonus A versus control and bonus B versus control individually using existing methods. Specifically, what is the motivation for decomposing the treatment into two stages, a base effect and an incremental effect, instead of modeling the treatments directly? Including an evaluation in RQ1 could help demonstrate the advantage of modeling multi-treatment effects in two stages compared to a direct treatment decomposition.

7. The notions of base effect and incremental effect appear to be misinterpreted. In fact, $\bar{\tau}^k(x_i)$ represents the overall treatment effect of assigning a bonus (averaging across bonuses A and B), whereas $\bar{\tau}_m^k(x_i)$ captures the deviation of the treatment effect for a specific bonus $m$ from this overall mean effect. This is also seen in Figure~4 (right panel), where $\bar{\tau}_m^k(x_i)$ values can be negative and positive, not necessarily incremental. Using the definition of CATE as an example, the left-hand side of the equation corresponds to the base effect, while the two expectations on the right-hand side represent the incremental effect. Note that the paper does not provide an explicit definition for these two terms, this formulation serves to illustrate the concept. Overall, I believe the interpretation of these two effects in the current presentation is somewhat misleading. Instead of base effect and incremental effect, it is more reasonable to call them overall effect and deviation from the overall effect for example, and change the interpretation in the main text.
    \begin{align*}
        \mathbb{E}(Y(1)-Y(0)\mid X)= \mathbb{E}(Y(1)-Y(0)\mid X, \hat{t}=1)\ p(\hat{t}=1\mid X) + \mathbb{E}(Y(1)-Y(0)\mid X, \hat{t}=0)\ p(\hat{t}=0\mid X).
    \end{align*}

8. Specifically, which $m$ is used in Figure~4 right panel.

9. In Figure 5, the x-axis is a discrete variable (bonus A vs. B), I'm confused why is the horizontal space filled rather than showing individual points? How should Figure~5 be interpreted in this context?

10. In Table 4, the MTMT (single-task) row, which specific task is used given that there are multiple tasks to choose from.

**Requested Changes:**

1. Provide explicit definitions for the estimand, and list identification assumptions.
2. Explicitly define $\bar{\tau}^k(x_i)$ and $\bar{\tau}_m^k(x_i)$ using counterfactual outcome language.
3. Correct/explain inconsistency in definitions in Equation~(1) and (8).
4. Explain intuition for Equation~(6).
5. Explain the motivation and rational for modeling multi-treatment as two stages rather than directly break down the treatment across different levels.
6. Correct intepretation/name for base effect and incremental effect.
7. Provide more details in experiments as listed in weakness 8,9, 10.

---

### Note · Authors · 2025-08-24

I have read and agree with the venue's withdrawal policy on behalf of myself and my co-authors.